# PMCHWT Solver Accelerated by Adaptive Cross Approximation for Efficient Computation of Scattering from Metal Nanoparticles

**DOI:** 10.3390/mi13071086

**Published:** 2022-07-08

**Authors:** Zhiwei Liu, Longfeng Xi, Yang Bao, Ziyue Cheng

**Affiliations:** 1Department of Information Engineering, East China Jiao Tong University, Nanchang 330029, China; xlfeng2021@163.com (L.X.); yue19960623@hotmail.com (Z.C.); 2College of Electronic and Optical Engineering, Nanjing University of Posts and Telecommunications, Nanjing 210042, China; 3State Key Laboratory of Millimeter Waves, Southeast University, Nanjing 210096, China

**Keywords:** PMCHWT, adaptive cross approximation, octree, metal nanoparticles, light scattering

## Abstract

An accelerated algorithm that can efficiently calculate the light scattering of a single metal nanoparticle was proposed. According to the equivalent principle, the method of moment (MoM) transforms the Poggio–Miller–Chang–Harrington–Wu–Tsai (PMCHWT) integral equations into linear algebraic equations, which are solved by the flexible generalized minimal residual solver (FGMRES). Each element of near field MoM impedance matrix was described by Rao–Wilton–Glisson (RWG) basis functions and calculated by double surface integrals. Due to the low-rank property, the adaptive cross approximation (ACA) algorithm based on the octree data structure was applied to compress the MoM impedance matrix of far field action leading to the significant reduction of solution time and memory. Numerical results demonstrated that the proposed method is both accurate and efficient. Compared with the traditional MoM, the ACA algorithm can significantly reduce the impedance matrix filling time and accelerate the scattering field’s computation from actual metal nanoparticles using PMCHWT integral equations.

## 1. Introduction

The unique optical properties of metal nanostructures have attracted significant attention from researchers in recent years [1,2]. In the optical band, the metal nanoparticles change from good conductor property to medium property. The physical mechanism of this phenomenon can be described by complex permittivity [3,4]. Surface plasmon polaritons (SPPs) are generated on the surface of the metal–dielectric interface when irradiated by light waves of a specific frequency [5]. For metal nanoparticles with a size smaller than or close to the wavelength, localized surface plasmon resonances (LSPRs) occur [6]. Theoretically, this phenomenon leads to three effects: enhanced light absorption, far-field scattering, and localized surface electric field. The functions and applications of optoelectronics, sensing, photoacoustic imaging, photothermal therapy, and detection based on the properties of metal nanostructured LSPRs have also been continuously developed, resulting in much meaningful progress in the application of LSPRs [7,8,9,10]. Therefore, discussing the rapid calculation of far-field scattering characteristics of metal nanoparticles can promote the research progress of LSPRs.

The study of the optical properties of metal LSPRs can be carried out using theoretical simulations and experiments. The theoretical simulations not only calculate the near-field and far-field optical properties of metal nanoparticles but also assist in designing experiments and theoretically analyzing the experimental results [11,12,13,14,15,16,17]. The optical response of metal can be well described in the classical framework based on Maxwell’s equations. Many numerical methods can be used for solving Maxwell’s equations, such as the finite difference time domain (FDTD) [18] method, finite element method (FEM) [19], and method of moment (MoM) [20]. FDTD and FEM are based on differential equations, while MoM is based on integral equations. When calculating metal scattering properties, FEM and FDTD need to discretize the scatterer and the surrounding space, while the MoM only needs to discretize the scatterer, thus reducing the time cost and storage. In addition, it is not easy to construct boundary conditions that describe open-domain problems. However, the Green function automatically satisfies the radiation boundary conditions, so MoM is used to calculate the scattering of any particle, ensuring high accuracy of the solution. For the electromagnetic problems of arbitrary three-dimensional targets, both the surface integral equation (SIE) [21] and the volume integral equation (VIE) [22] can be applied. Since the metal nanoparticle is considered a uniform medium, the SIE is preferred with fewer unknowns than VIE. Reference [23] used the SIE and VIE formulas to calculate the bistatic scattering of the plasma, respectively and then compared with the accurate results to show the superiority of the SIE method.

The major computational costs of the MoM for solving SIEs are combined in two parts: they are the times for filling the impedance matrix and iteratively solving the equations. According to reference [12], the most accurate and reliable formula among the SIEs commonly used in plasma simulation is PMCHWT integral equations [24,25]. However, the diagonal elements of the impedance matrix established by the PMCHWT integral equations are not dominant, and the iterative solution suffers from the issue of slow convergence. This deficiency can be overcome if preprocessing is performed before the iterative solution [12]. In reference [26], FGMRES was considered to replace GMRES for solving large linear equations. There is a preprocessing technique based on the traditional GMRES solver added to the FGMRES solver, which makes the iteration convergence more efficient. By reducing the number of steps required for convergence, FGMRES achieves the purpose of saving calculation time.

On the other hand, the accelerated impedance matrix filling can be carried out in two ways. One is to reduce the calculation amount of unknowns by improving the calculation accuracy of matrix elements [27], and the other is to reduce the calculation time by using the additional algorithm to accelerate the calculation of matrix elements without reducing the unknowns. This article chose the second method. ACA [28] is a purely algebraic acceleration algorithm based on octree structure grouping and low-rank matrix compression, which can speed up impedance matrix filling. An octree is a three-dimensional spatial hierarchical tree-like data structure, which effectively represents the grouping of the far and near fields of three-dimensional objects. The impedance matrix formed by the far-field region has a low-rank characteristic, which utilizes the decomposition and compression characteristics of ACA to speed up the filling of the impedance matrix.

The ACA algorithm was proposed by Bebendorf in 2000 [28]. It uses the mathematical method to process the matrix and decreases the computational complexity and memory requirements. In 2005, Zhao introduced the ACA algorithm into the MoM to analyze electromagnetic radiation and scattering problems [29]. Since the impedance matrix is composed of the interaction between the field point and the source point, the farther the distance between the field point and the source point is, the smaller the effect is, which is reflected in the impedance matrix in that the far-field group matrix block is low rank. The ACA algorithm uses the low-rank characteristic of the far-field group matrix block to decompose and compress it to accelerate the filling of the impedance matrix.

Gold and silver are commonly studied in metals. Gold nanospheres have plasmon resonances in the region where gold is highly absorbing. The resonances are therefore strongly damped, but gold spheroids or silver nanospheres at resonance may prove more difficult. This paper focused on the calculation of bistatic RCS of a single metal nanoparticle (taking the calculation of a gold nanoparticle as an example). The electromagnetic scattering characteristics of metal nanoparticles were analyzed based on the PMCHWT integral equations by using the MoM as the numerical solution method. Combined with the ACA algorithm of matrix compression, the fast filling of the impedance matrix was accelerated to realize the fast calculation of far-field scattering characteristics of metal nanoparticles.

## 2. Materials and Methods

### 2.1. Generic PMCHWT Formulation for a Metal Nanoparticle

Considering a metal nanoparticle in a homogeneous medium space, the subscripts of the metal nanoparticle, the homogeneous medium, and the free space related parameters are represented by 1, 2, and 0, respectively. *ε**_1_* = *ε_1r_ε*_0_ and *μ_1_* = *μ_1r_μ*_0_ are the complex permittivity and permeability of the metal nanoparticle; *ε_2_* = *ε_2r_ε*_0_ and *μ_2_ = μ_2r_μ_0_* are the complex permittivity and permeability of the homogeneous medium; *ε*_0_ and *μ_0_* are the permittivity and permeability of free space. Where *ε_ir_*, *μ_ir_, i = 1,2* are the relative permittivity and relative permeability. The surface of the metal nanoparticle is denoted as *S*. There is uniform plane wave (**E***^inc^*, **H***^inc^*) irradiation outside the surface *S.* In the internal and external regions of the metal nanoparticle, the equivalent model is constructed by using the equivalent principle to solve the distribution of internal and external fields. The equivalent model is shown in Figure 1.

Where $\hat{n}$ and $\hat{t}$ are the unit normal and unit tangential vector of the outer surface of the medium, respectively; **J***_e_* and **J***_m_* are the equivalent currents and magnetic currents on the outer surface, respectively; −**J***_e_* and −**J***_m_* are the equivalent currents and magnetic currents on the inner surface, respectively; $E_{1}^{sca}$ and $H_{1}^{sca}$ are the scattering electric field and the scattering magnetic field of the inner surface, respectively; $E_{2}^{sca}$ and $H_{2}^{sca}$ are the scattering electric field and the scattering magnetic field of the outer surface, respectively.

Equations (1) and (2) calculate the scattering electric and magnetic fields, respectively:(1)$$E_{i}^{sca}=-\gamma_{i}\eta_{i}\int_{S}{\overset{=}{G}}_{i}\left(r,r^{\prime }\right)\cdot J_{e}\left(r^{\prime }\right)dS-\int_{S}\nabla G_{i}\left(r,r^{\prime }\right)\times J_{m}\left(r^{\prime }\right)dS,\;i=1,2$$
(2)$$H_{i}^{sca}=-\frac{\gamma_{i}}{\eta_{i}}\int_{S}{\overset{=}{G}}_{i}\left(r,r^{\prime }\right)\cdot J_{m}\left(r^{\prime }\right)dS+\int_{S}\nabla G_{i}\left(r,r\prime \right)\times J_{e}\left(r^{\prime }\right)dS,\;i=1,2$$ where *γ_1_* and *γ_2_* represent the propagation constants of the inner and outer regions, respectively; Equation (3) defines the propagation constants; *η_1_* and *η_2_* represent the wave impedance in the inner and outer regions, respectively; Equation (4) defines the wave impedance; *P.V.* represents the principal value integral; Meanwhile, $G_{i}$ and ${\overset{=}{G}}_{i}$ represent the corresponding regions of Green function and dyadic Green function, respectively; Equation (5) defines the Green function, and Equation (6) defines the dyadic Green function.
(3)$$\gamma_{i}=\omega \sqrt{\mu_{i}\varepsilon_{i}},\;i=1,2$$
(4)$$\eta_{i}=\sqrt{\frac{\mu_{i}}{\varepsilon_{i}}},\;i=1,2$$
(5)$$G_{i}\left(r,r^{\prime }\right)=\frac{e^{-\gamma_{i}\left|r-r^{\prime }\right|}}{4\pi \left|r-r^{\prime }\right|},\;i=1,2$$
(6)$${\overset{=}{G}}_{i}\left(r,r^{\prime }\right)=\left(\overset{=}{I}-\frac{\nabla \nabla }{\gamma_{i}{}^{2}}\right)G_{i}\left(r,r^{\prime }\right),\;i=1,2$$
where **r**’ and **r** represent the source point and observation point, respectively. Equation (7) introduces boundary conditions of electric field and magnetic field on the outer surface; Equation (8) introduces boundary conditions of the electric field and magnetic field on the inner surface:(7)$$\hat{n}\times \left(E^{inc}+E_{2}^{sca}\right)=-J_{m},\;\hat{n}\times \left(H^{inc}+H_{2}^{sca}\right)=J_{e}$$
(8)$$\hat{n}\times E_{1}^{sca}=J_{m},\;\hat{n}\times H_{1}^{sca}=-J_{e}$$

Then, we establish normal surface integral Equations (9) and (10) for electric current and magnetic current by combining the scattered electromagnetic fields and boundary conditions on the inner and outer surfaces, respectively:(9)$$\hat{n}\times E^{inc}\left(r\right)=\sum_{i=1}^{2}\hat{n}\times \gamma_{i}\eta_{i}\int_{S}{\overset{=}{G}}_{i}\left(r,r^{\prime }\right)\cdot J_{e}\left(r^{\prime }\right)dS+\hat{n}\times P.V.\int_{S}\nabla G_{i}\left(r,r^{\prime }\right)\times J_{m}\left(r^{\prime }\right)dS$$
(10)$$\hat{n}\times H^{inc}\left(r\right)=\sum_{i=1}^{2}\hat{n}\times \frac{\gamma_{i}}{\eta_{i}}\int_{S}{\overset{=}{G}}_{i}\left(r,r^{\prime }\right)\cdot J_{m}\left(r^{\prime }\right)dS-\hat{n}\times P.V.\int_{S}\nabla G_{i}\left(r,r^{\prime }\right)\times J_{e}\left(r^{\prime }\right)dS$$

Next, the tangential PMCHWT Equations (11) and (12) are obtained by applying $\hat{n}\times$ on both sides of Equations (9) and (10):(11)$$\hat{t}\hat{t}\cdot E^{inc}\left(r\right)=\sum_{i=1}^{2}\hat{t}\hat{t}\cdot \gamma_{i}\eta_{i}\int_{S}{\overset{=}{G}}_{i}\left(r,r^{\prime }\right)\cdot J_{e}\left(r^{\prime }\right)dS+\hat{t}\hat{t}\cdot P.V.\int_{S}\nabla G_{i}\left(r,r^{\prime }\right)\times J_{m}\left(r^{\prime }\right)dS$$
(12)$$\hat{t}\hat{t}\cdot H^{inc}\left(r\right)=\sum_{i=1}^{2}\hat{t}\hat{t}\cdot \frac{\gamma_{i}}{\eta_{i}}\int_{S}{\overset{=}{G}}_{i}\left(r,r^{\prime }\right)\cdot J_{m}\left(r^{\prime }\right)dS-\hat{t}\hat{t}\cdot P.V.\int_{S}\nabla G_{i}\left(r,r^{\prime }\right)\times J_{e}\left(r^{\prime }\right)dS$$

Finally, balancing the magnitudes of the electric and magnetic fields of Equations (11) and (12) with the wave impedance *η*_0_, yielding the tangential PMCHWT Equations (13) and (14):(13)$$\hat{t}\hat{t}\cdot E^{inc}\left(r\right)=\sum_{i=1}^{2}\hat{t}\hat{t}\cdot \gamma_{i}\eta_{ir}\int_{S}{\overset{=}{G}}_{i}\left(r,r^{\prime }\right)\cdot {\tilde{J}}_{e}\left(r^{\prime }\right)dS+\hat{t}\hat{t}\cdot P.V.\int_{S}\nabla G_{i}\left(r,r^{\prime }\right)\times J_{m}\left(r^{\prime }\right)dS$$
(14)$$\hat{t}\hat{t}\cdot \eta_{0}H^{inc}\left(r\right)=\sum_{i=1}^{2}\hat{t}\hat{t}\cdot \frac{\gamma_{i}}{\eta_{ir}}\int_{S}{\overset{=}{G}}_{i}\left(r,r^{\prime }\right)\cdot J_{m}\left(r^{\prime }\right)dS-\hat{t}\hat{t}\cdot P.V.\int_{S}\nabla G_{i}\left(r,r^{\prime }\right)\times {\tilde{J}}_{e}\left(r^{\prime }\right)dS$$
where,${\;\eta }_{\mathrm{ir}}=\frac{\eta_{i}}{\eta_{0}}$, *i* = 1,2, ${\tilde{J}}_{e}\left(r’\right){=\eta }_{0}J_{e}\left(r’\right)$.

The current density and magnetic current density of the electric field integral equation are expanded by the RWG basis function **Λ**_n_(**r**’):(15)$${\tilde{J}}_{e}\left(r^{\prime }\right)=\eta_{0}J_{e}\left(r^{\prime }\right)=\sum_{n=1}^{N}\alpha_{n}\Lambda_{n}\left(r^{\prime }\right),\;J_{m}\left(r^{\prime }\right)=\sum_{n=1}^{N}\beta_{n}\Lambda_{n}\left(r^{\prime }\right)$$

When the unknown functions of the PMCHWT integral equations are expanded by the basis function **Λ***_n_*(**r**’), the task of solving the integral equations is transformed into the unknown coefficients *α_n_* and *β_n_* of the basis function. The number of expansion basis functions is the number of equations unknowns.

When the MoM is used to analyze the SIE of a target, the surface of any shape object can be simulated by a planar triangular patch, and the surface current and magnetic current can be described by a planar RWG basis function. Each RWG basis function is composed of two adjacent triangles, describing the current or magnetic current flowing from one triangle (positive triangle) to another (negative triangle). The mathematical expression is as in Equation (16):(16)$$\Lambda_{n}\left(r\right)=\left\{\begin{array}{cc} \frac{l_{n}}{2A_{n}^{+}}\rho_{n}^{+} & r\in T_{n}^{+}\\ \frac{l_{n}}{2A_{n}^{-}}\rho_{n}^{-} & r\in T_{n}^{-}\\ 0 & others \end{array}\right.$$
where $A_{n}^{\pm }$ represents the area of the corresponding triangle, *l_n_* is the side length, $\rho_{n}^{\pm }$ is the direction vector of the current in the triangle, and $T_{n}^{\pm }$ represents the positive and negative triangle. The definitions of each symbol are shown in Figure 2.

The RWG basis function has two important properties. The first characteristic is the continuity of the normal component of the edge, which ensures the continuity of the current across the common edge; the second characteristic is that the divergences of the basis function of the two triangles are equal, and the symbols are opposite, which ensures that the total charge corresponding to the basis function is zero. Its expression is as in Equation (17):(17)$$\nabla \cdot \Lambda_{n}\left(r\right)=\left\{\begin{array}{cc} \frac{l_{n}}{A_{n}^{+}} & r\in T_{n}^{+}\\ -\frac{l_{n}}{A_{n}^{-}} & r\in T_{n}^{-}\\ 0 & others \end{array}\right.$$

Since the RWG basis function is a vector basis function, it is often used to expand the surface current density and magnetic current density.

The structure of the impedance matrix of the PMCHWT equations is shown in Figure 3.

The impedance matrix elements are calculated by Equations (18)–(21). The double integrals $\int_{S_{m}}^{\;}\Lambda_{m}\left(r\right)\cdot \int_{S_{n}}^{\;}{\overset{=}{G}}_{i}\left(r,r^{\prime }\right)\cdot \Lambda_{n}\left(r^{\prime }\right)dS^{\prime }dS$ and $\int_{S_{m}}^{\;}\Lambda_{m}\left(r\right)\cdot \int_{S_{n}}^{\;}\nabla G_{i}\left(r,r^{\prime }\right)\times \Lambda_{n}\left(r^{\prime }\right)dS^{\prime }dS$ involved in $Z_{mn}^{EJ}$, $Z_{mn}^{EM}$, $Z_{mn}^{HJ}$, and $Z_{mn}^{HM}$ are calculated by six-point Gaussian integral.
(18)$$Z_{mn}^{EJ}=\sum_{i=1}^{2}\gamma_{i}\eta_{ir}\int_{S_{m}}\Lambda_{m}\left(r\right)\cdot \int_{S_{n}}{\overset{=}{G}}_{i}\left(r,r^{\prime }\right)\cdot \Lambda_{n}\left(r^{\prime }\right)dS^{\prime }dS$$
(19)$$Z_{mn}^{EM}=\sum_{i=1}^{2}\int_{S_{m}}\Lambda_{m}\left(r\right)\cdot \int_{S_{n}}\nabla G_{i}\left(r,r^{\prime }\right)\times \Lambda_{n}\left(r^{\prime }\right)dS^{\prime }dS$$
(20)$$Z_{mn}^{HJ}=-\sum_{i=1}^{2}\int_{S_{m}}\Lambda_{m}\left(r\right)\cdot \int_{S_{n}}\nabla G_{i}\left(r,r^{\prime }\right)\times \Lambda_{n}\left(r^{\prime }\right)dS$$
(21)$$Z_{mn}^{HM}=\sum_{i=1}^{2}\frac{\gamma_{i}}{\eta_{ir}}\int_{S_{m}}\Lambda_{m}\left(r\right)\cdot \int_{S_{n}}{\overset{=}{G}}_{i}\left(r,r^{\prime }\right)\cdot \Lambda_{n}\left(r^{\prime }\right)dS$$

The right vector elements are calculated by Equations (22) and (23):(22)$$V_{m}^{E}=\int_{S_{m}}\Lambda_{m}\left(r\right)\cdot E^{inc}\left(r\right)dS$$
(23)$$V_{m}^{H}=\eta_{0}\int_{S_{m}}\Lambda_{m}\left(r\right)\cdot H^{inc}\left(r\right)dS$$

### 2.2. Triangular–Triangular Cyclic Integral Method for Element Calculation of Impedance Matrix

In the calculation of the traditional MoM, the cycle unit filled by impedance matrix elements is often the basis function itself. Each current RWG basis function contains two triangular patches for the commonly used triangulation. It is easy to find that the basis functions of non-single-sided basis function units such as RWG as they often have overlapping regions in the solution region, which leads to a large number of repeated calculations in the process of taking the basis function itself as the iterative unit. If the RWG–RWG cyclic integral is used, three edges of each triangle correspond to an RWG basis function. The same triangle area score will be repeated nine times in the iterative integral of the interaction of a pair of basis functions. If the triangle–triangle cyclic integral is used instead of the RWG–RWG, the repeated calculation can be avoided, which dramatically improves the calculation speed and iterative efficiency of matrix elements.

### 2.3. The Octree Establishes Grouping

The ACA algorithm is based on the low-rank matrix factorization principle, so it is applied to solve the matrix equation in the electromagnetic field and must have certain preconditions; that is, it can only be used to solve the far-field impedance matrix generated by far-field interaction. The impedance matrix generated by near-field interaction is generally of high rank, which is suitable for solving by the traditional MoM. Therefore, the basis function must be grouped. To make the established matrix have good performance, in the regional division of the whole three-dimensional object, we should try to ensure that each sub-domain has little difference. An octree is a tree-like data structure used to describe three-dimensional space. The basic idea of the subdivision is equalizing in every direction. In the specific subdivision process, we use a cube to represent the space where the entity is located; that is, the octree starts from a cube. The first divides it into eight sub-cubes, the next continues to decompose according to certain rules, and then divides some sub-cubes into eight sub-cubes, according to this recursive division, until a certain division termination condition is satisfied. The whole partitioning process can be represented by a tree structure with 8 or 0 child nodes per node.

In this paper, the metal nanoparticle was stratified by a square box. The square’s side length is the maximum length of the metal nanoparticle in the three coordinate axes of the three-dimensional coordinate system. The specific octree structure can be defined in the following way.

As shown in Figure 4, the cube surrounding the entire metal nanoparticle is first divided into eight small cubes; because these eight cubes cannot distinguish between far interaction or near interaction, each cube has to be further divided down. Therefore, there are 64 small cubes in the second division. We take these 64 cubes divided twice as the first layer and then divide them again as the second layer. In this way, this is the whole process of octree division of the object region. After the partition, the square boxes in the whole region are divided into non-empty groups and empty groups according to whether the small square contains the split triangular mesh. Empty groups are groups that do not need to be calculated and are discarded directly. According to the distance between non-empty boxes, the interaction is divided into far-field interaction and near-field interaction, and then calculated, respectively.

The simplified two-dimensional plane diagram is shown in Figure 5. The self-acting group refers to the interaction between the basis functions in the same square box; the near-field group refers to the interaction between adjacent square boxes; the far-field group refers to the interaction between square boxes that are not separated.

### 2.4. Overview of the Adaptive Cross Approximation (ACA) Algorithm 

The ACA algorithm draws on the octree structure model, divides the basis functions into each group according to the vertex position, then divides the impedance matrix into many sub-matrix blocks with different dimensions. The diagonal matrix block is a full rank matrix, representing the interaction between self-acting groups and near-field groups. The traditional MoM is used to calculate. The non-diagonal matrix block is a low-rank matrix block that represents the interaction between far-field groups and is calculated by the ACA algorithm. The matrix block diagram is shown in Figure 6.

The ACA algorithm is a pure algebraic algorithm, which approximately decomposes a large matrix into the multiplication of two small matrices according to the decomposition principle of matrices [30]. Assuming that the rank of the matrix **A** is *r*, the number of dimension objective basis functions of the matrix is determined, and it is assumed to be *m*×*n*, which is approximately decomposed into the multiplication of two matrices **U** and **V**. Matrix decomposition is shown in Figure 7, and its expression in Equation (24):(24)$$A^{m\times n}=U^{m\times r}\cdot V^{r\times n}$$

The rank of a matrix can be artificially set by stopping criteria. Equation (25) defines the threshold condition of iteration.
(25)$$\left\|{\tilde{R}}^{m\times n}\right\|=\left\|A^{m\times n}-{\tilde{A}}^{m\times n}\right\|\leq \varepsilon \left\|A^{m\times n}\right\|$$
where **R** is an error matrix, *ε* denotes the stop precision, and ‖·‖ means the matrix Frobenius specification.

The detailed process of the ACA Algorithm 1 is as follows [29]:
**Algorithm 1. The detailed process of the ACA algorithm**
Initialization steps:
(1)Initializing the index of the first row: *I*_1_ = 1, suppose $\tilde{A\;}=0$;(2)Initializing the first row of the error matrix: $\tilde{R}\left(I_{1},:\right)=A\left(I_{1},:\right)$;(3)Determining the index *J*_1_ of the first column: $\left|\tilde{R}\left(I_{1}{,J}_{1}\right)\right|{=\max}_{j}\left(\left|\tilde{R}\left(I_{1},j\right)\right|\right)$;
(4)Obtaining the first row of the **V** matrix: $v_{1}=\tilde{\;R}\left(I_{1},:\right)/\left|\tilde{R}\left(I_{1}{,J}_{1}\right)\right|$;(5)Initializing the first column of the error matrix: $\tilde{R}\left({:,J}_{1}\right)=A\left({:,J}_{1}\right)$;(6)Obtaining the first column of the **U** matrix: $u_{1}=\tilde{R}\left({:,J}_{1}\right)$;(7)Computing the approximate matrix: $\|{\tilde{A}}^{\left(1\right)}{\|}^{2}{=\|\tilde{A}}^{\left(0\right)}{\|}^{2}{+\|u}_{1}{\|}^{2}\|v_{1}{\|}^{2}$;(8)Determining the index of the second row *I*_2_: $\left|\tilde{R}\left(I_{2}{,J}_{1}\right)\right|{=\max}_{j}\left(\left|\tilde{R}\left({i,J}_{1}\right)\right|\right)$;

Next, the *k* th iteration:
(1)Updating the *I_k_* th row of the approximation matrix of the error:$\tilde{R}\left(I_{k},:\right)=A\left(I_{k},:\right)-\sum_{l=1}^{k-1}{\left(u_{l}\right)}_{I_{k}}v_{l}$;(2)Finding the maximum value in line *I_k_* to determine the *J_k_* th column:$\left|\tilde{R}\left(I_{k}{,J}_{k}\right)\right|{=\max}_{j}\left(\left|\tilde{R}\left(I_{k},j\right)\right|\right)$, *j* ≠ *J*_1_,…,*J_k_*;(3)Obtaining the *k* th row of the **V** matrix: $v_{k}=\tilde{\;R}\left(I_{k},:\right)$;(4)Updating the *J_k_* th column of the error matrix: $\tilde{R}\left({:,J}_{k}\right)=A\left({:,J}_{k}\right)\;-\;\sum_{l=1}^{k-1}{\left(v_{l}\right)}_{J_{k}}u_{l}$;(5)Getting the *k* th column of the **U** matrix: $u_{k}=\tilde{\;R}\left({:,J}_{k}\right)/\left|\tilde{R}\left(I_{k}{,J}_{k}\right)\right|$;(6)Computing the approximate matrix:$\|{\tilde{A}}^{\left(k\right)}{\|}^{2}=\|{\tilde{A}}^{\left(k-1\right)}{\|}^{2}+2\sum_{j=1}^{k-1}\left|u_{j}^{T}u_{k}\right|\;\cdot \;\left|v_{j}^{T}v_{k}\right|\|u_{k}{\|}^{2}\|v_{k}{\|}^{2}$;(7)Judging convergence error, if $\left\|u_{k}\|\|v_{k}\|\;\leq {\;\varepsilon \;\tilde{A}}^{\left(k\right)}\right\|$, end iteration, else go on;(8)Continue looking for the next line *I_k_*_+1_: $\left|\tilde{R}\left(I_{k+1}{,J}_{k}\right)\right|{=\max}_{j}\left(\left|\tilde{R}\left({i,J}_{k}\right)\right|\right)$, *i* ≠ *I*_1_,…,*I_k_*.


When the ACA algorithm is used to solve the matrix **U** and **V**, according to the adaptive principle, only *k* rows and *k* columns in the matrix are selected, and each element of the impedance matrix is not calculated in advance, which reduces the complexity. Since *k* < min(*m*,*n*), the calculation number of impedance matrix elements is *k* × (*m* + *n*), far less than *m* × *n*, and the computational complexity is reduced to 2*kX*, where *X* is the number of unknowns.

The most prominent feature of the ACA algorithm is that it does not depend on Green’s function and integral equation in the solution process. In addition, another advantage of the ACA algorithm is that it approximates the original matrix only by knowing the elements of the original matrix and updating the specific column information.

### 2.5. ACA Algorithm Accelerates Filling Impedance Matrix 

The octree data structure has divided all unknowns into near-field and far-field groups. The near-field group’s impedance matrix filling adopts the traditional storage method, and all impedance elements are stored in sequence. The far-field impedance matrix elements are screened by the ACA algorithm, and the sparse storage method only needs to store *k* rows and *k* columns of elements, which significantly saves memory. At the same time, the ACA algorithm changes the matrix–vector multiplication order for the iterative solution of the equation. As shown in Figure 8, $A^{n\times n}\cdot X^{n}=b^{n}$ in the traditional MoM is changed to $U^{n\times k}\cdot \left(V^{k\times n}\cdot X^{n}\right)=b^{n}$, where *n* is the number of unknowns. The amount of calculation is changed from *n* × *n* to 2*kn*, plus the element filling time of the impedance matrix of the near-field group, and the overall time complexity is reduced to *n*log*n*.

## 3. Results and Discussion

In this section, we provide some examples to prove the accuracy and efficiency of the PMCHWT integral equations with the ACA algorithm in analyzing the far-field scattering characteristics of a metal nanoparticle. The CPU model used in all examples in this section is Intel CoreTMi5-7500, and the memory capacity is 8 GB. The relative permeability is 1.0.

Considering a gold nanosphere with a radius of 200 nm analyzed in reference [14], the uniform plane wave with a wavelength of 550 nm is illuminated on the metal nanosphere; meanwhile, the pitch angle is 0°, and the azimuth angle is 0°. The relative dielectric constant of the gold nanosphere at this incident wavelength is listed in reference [3] as *ε_1r_* = −8.0 − *j*1.66. The number of unknowns is 1920. Figure 9a calculates the HH polarization and VV polarization of the bistatic RCS of the metal sphere; Figure 9b–d calculate the memory consumption of the near-field impedance matrix, the impedance matrix filling time, and the time required for each iteration step respectively after applying the ACA algorithm under HH polarization.

It is obtained from Figure 9a–d that the PMCHWT equations solved by the pure moment method and the PMCHWT equations solved by the ACA algorithm acceleration are in good agreement and also reduce the computation time from *N*^2^ to *N*log(*N*), where N is the number of unknowns. Table 1 shows the time comparison between filling impedance matrix with pure MoM and accelerating the filling impedance matrix with the ACA algorithm.

Considering a gold nanosphere with a radius of 68.25 nm analyzed in reference [15] and a radius of 456 nm analyzed in reference [17], respectively, the uniform plane wave with a wavelength of 546 nm is illuminated on the metal nanosphere; meanwhile, the pitch angle is 0°, and the azimuth angle is 0°. The relative dielectric constant of the gold nanosphere at this incident wavelength is listed in reference [3] as *ε_1r_* = −5.84 − *j*2.11. The number of unknowns is 1920. Figure 10a,b calculate the HH polarization and VV polarization of the bistatic RCS of the metal sphere. It can be seen from the graph that the PMCHWT equations solved by the pure moment method and the PMCHWT equations solved by the ACA algorithm acceleration are in good agreement. Table 2 shows the time comparison between the filling impedance matrix with pure moment method and accelerating the filling impedance matrix with ACA algorithm.

## 4. Conclusions

A method for rapidly calculating the light scattering of a metal nanoparticle was proposed. Aiming at the light-scattering problem of metal nanoparticles, the MoM discretized PMCHWT integral equations, and, finally, the integral equation system was solved by FGMRES. Due to the low-rank characteristic of the far-field impedance matrix, a pure algebraic ACA algorithm was used to compress the far-field impedance matrix, which reduced the computation time. Numerical examples verified the effectiveness and versatility of the ACA-based PMCHWT integral equations for solving metal nanoparticles. Specifically, three examples in the literature were considered. Through these examples, it was concluded that when the number of unknowns remains unchanged, the ACA algorithm significantly reduces the filling time of the impedance matrix compared with the traditional MoM.

## Figures and Tables

**Figure 1 micromachines-13-01086-f001:**
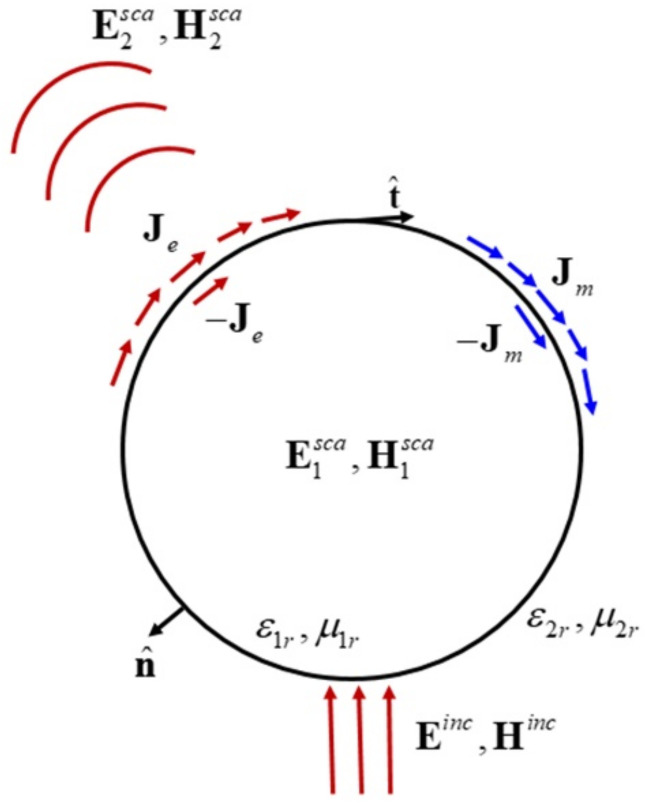
Scattering model of a spherical metal nanoparticle.

**Figure 2 micromachines-13-01086-f002:**
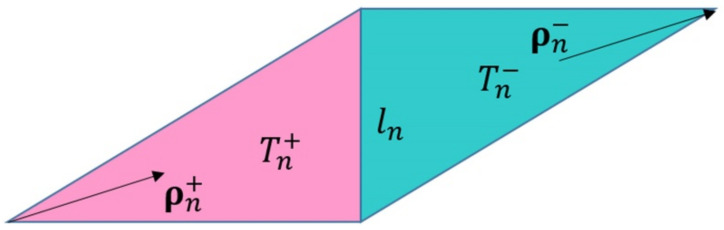
Planar RWG basis function diagram.

**Figure 3 micromachines-13-01086-f003:**
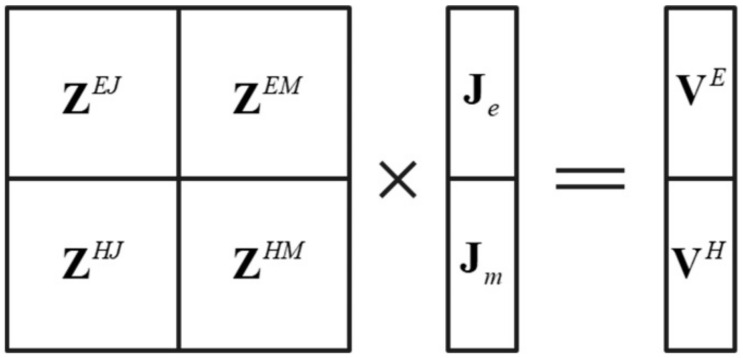
The structure of the impedance matrix of the PMCHWT equations.

**Figure 4 micromachines-13-01086-f004:**
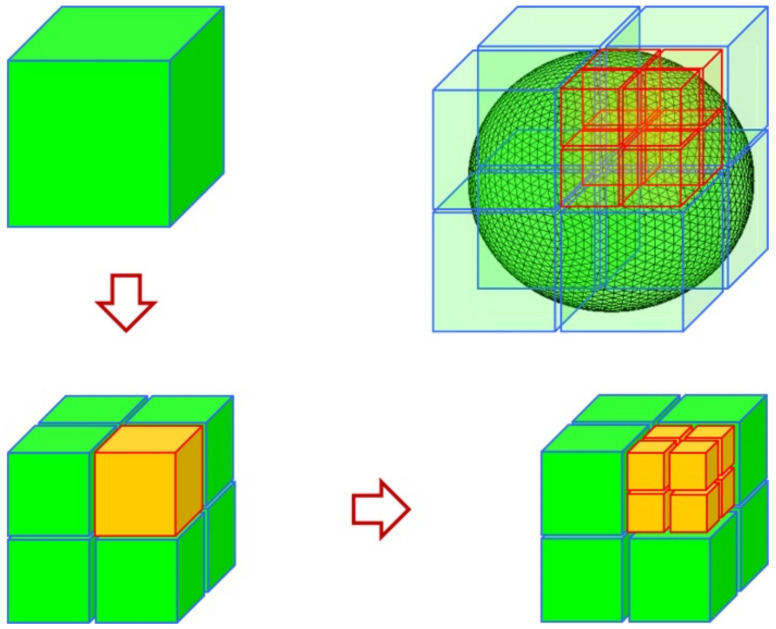
A layer of a tree structure.

**Figure 5 micromachines-13-01086-f005:**
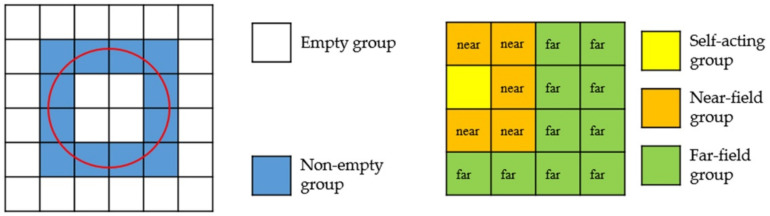
The far and near field diagram of non-empty group division.

**Figure 6 micromachines-13-01086-f006:**
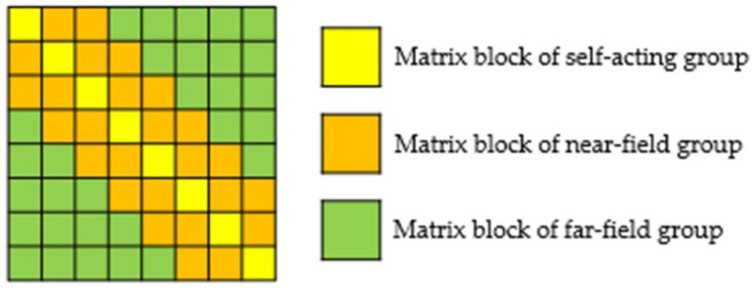
Matrix block diagram.

**Figure 7 micromachines-13-01086-f007:**
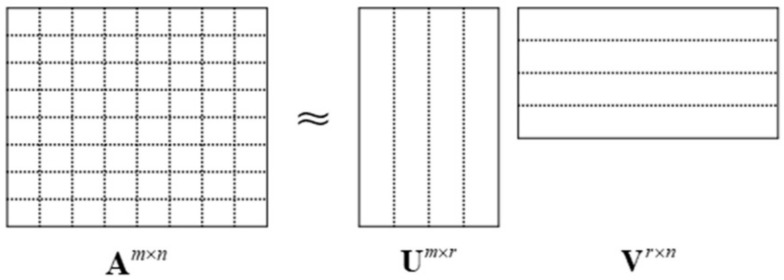
Matrix **A** compression approximate decomposition diagram.

**Figure 8 micromachines-13-01086-f008:**
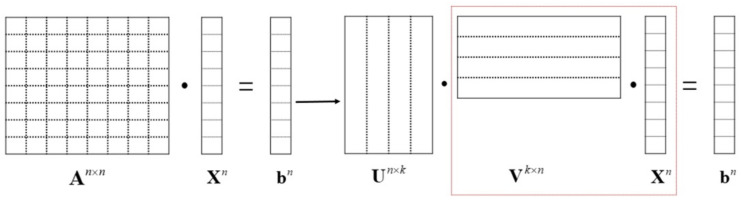
Matrix–vector multiplication order changed using the ACA algorithm.

**Figure 9 micromachines-13-01086-f009:**
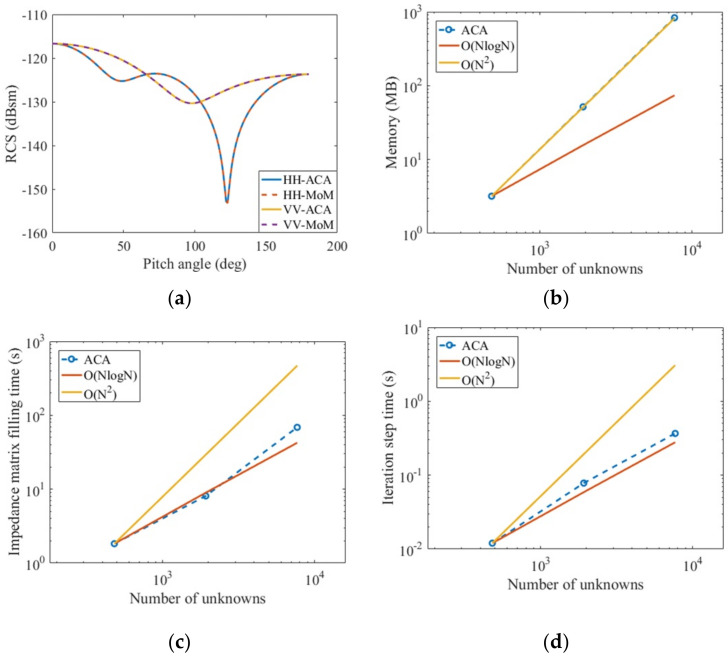
Related analysis of a gold nanosphere calculated by MoM and ACA: radius 200 nm, incident wavelength 550 nm, *ε_1r_* = −8.0 – *j* 1.66, *μ_1r_* = 1.0. (**a**) Bistatic RCS; (**b**) memory occupied by the near field matrix; (**c**) filling time of impedance matrix; (**d**) average solution time per iteration.

**Figure 10 micromachines-13-01086-f010:**
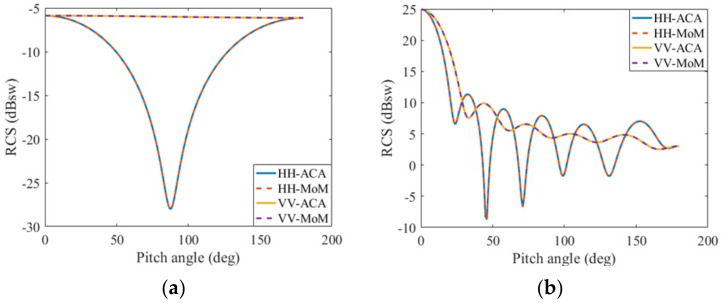
Bistatic RCS of a gold nanosphere calculated by MoM and ACA: incident wavelength 546 nm, *ε_1r_* = −5.84 − *j*2.11, *μ_1r_* = 1.0. (**a**) radius 68.25 nm; (**b**) radius 546 nm.

**Table 1 micromachines-13-01086-t001:** Time comparison between MoM and ACA calculation with radius 200 nm.

Method	MoM	ACA
HH polarization calculation time (s)	42.64	28.84
VV polarization calculation time (s)	42.61	28.86

**Table 2 micromachines-13-01086-t002:** Time comparison between MoM and ACA calculation with radius 68.25 nm and 456 nm.

Radius (nm)	68.25		456	
Method	MoM	ACA	MoM	ACA
HH polarization calculation time (s)	41.30	27.90	45.22	30.37
VV polarization calculation time (s)	41.19	27.47	44.61	30.57

## Data Availability

Not applicable.

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
