# Peer review of "PMCHWT Solver Accelerated by Adaptive Cross Approximation for Efficient Computation of Scattering from Metal Nanoparticles"

_micromachines, 2022, doi:10.3390/mi13071086_

Round 1

Reviewer 1 Report

MoM is a numerical calculation method of electromagnetic field based on integral Maxwell’s equation. ACA is a pure algebraic acceleration algorithm based on octree structure grouping and low-rank matrix compression, which can speed up the filling of the impedance matrix. Therefore, the ACA algorithm applied to MOM can accelerate the calculation of far-field scattering characteristics of metal nanoparticles. Numerical examples are given to prove the accuracy and effectiveness of the proposed algorithm

The paper shows the detailed and rigorous derivation process. The whole paper, with rigorous structure and smooth logic, has some practical significance and is recommended to be published.

Questions:

1.        What beam is the incident field used in the numerical simulation?

2.        On page3, Please specify the meaning of G in Equations (1a) and (1b).

Reviewer 2 Report

This manuscript demonstrates how the use of the Adaptive Cross Approximation (ACA) algorithm based on an octree data structure can speed up Method of Moment (MoM) calculations.

The method (originally described in Ref. 27) is explained in detail and then demonstrated in the special case of gold nanoparticles within the PMCHWT formulation of the MoM.

This work is of interest to researchers studying or using the MoM.

There are however few minor issues that should be addressed:

- The English is not always correct in some parts, which makes some sentences hard to read/understand. A careful proofreading would be beneficial.

- Most of the references about applying MoM in metallic nanoparticles are from a single group. It would be good to include work from other groups, including some earlier one, for example Kern et al., JOSA 26, 732 (2009), Raziman et al, JOSAB 32, 485 (2015).

- There is no detail about how the integrals are calculated to fill the impedance matrix (quadrature scheme, any analytic treatment). The calculation approach is likely to be the most important factor in determining the computing time to fill in the impedance matrix.

- As mentioned in the introduction, gold nanoparticles are commonly used in plasmonics applications, typically for their plasmon resonances. However, all the examples provided by the author relate to single-wavelength properties (such as the bistatic RCS). I would be good to discuss at least one example of a computed resonance spectrum with the standard and accelerated methods.

- Gold nanospheres have plasmon resonances in the region where gold is very absorbing. The resonances are therefore strongly damped and one could argue that this could be an "easier" case to solve. Gold spheroids or silver nanospheres at resonance may prove more difficult. If such studies are not possible, this drawback should at least be mentioned.

Once these issues are addressed, I recommend publication in Micromachines.
